# O-RADS Classification for Ultrasound Assessment of Adnexal Masses: Agreement between IOTA Lexicon and ADNEX Model for Assigning Risk Group

**DOI:** 10.3390/diagnostics13040673

**Published:** 2023-02-10

**Authors:** Julio Vara, Mariachiara Pagliuca, Serena Springer, Juan Gonzalez de Canales, Isabel Brotons, Javiera Yakcich, Silvia Ajossa, Maria Angela Pascual, Stefano Guerriero, Juan Luis Alcazar

**Affiliations:** 1Department of Obstetrics and Gynecology, Clinica Universidad de Navarra, University of Navarra, 31008 Pamplona, Spain; 2Centro Integrato di Procreazione Medicalmente Assistita (PMA) e Diagnostica Ostetrico-Ginecologica, Azienda Ospedaliero Universitaria-Policlinico Duilio Casula Monserrato, University of Cagliari, 09124 Cagliari, Italy; 3Department of Medical and Surgical Sciences, Universita degli Studi di Trieste, 34127 Trieste, Italy; 4Faculty of Medicine, Los Andes University, Santiago 12455, Chile; 5Department of Obstetrics, Gynecology and Reproduction, Institut Universitari Dexeus, 08028 Barcelona, Spain

**Keywords:** adnexal mass, ultrasound, diagnosis, classification, O-RADS

## Abstract

Background: The O-RADS system is a new proposal for establishing the risk of malignancy of adnexal masses using ultrasound. The objective of this study is to assess the agreement and diagnostic performance of O-RADS when using the IOTA lexicon or ADNEX model for assigning the O-RADS risk group. Methods: Retrospective analysis of prospectively collected data. All women diagnosed as having an adnexal mass underwent transvaginal/transabdominal ultrasound. Adnexal masses were classified according to the O-RADS classification, using the criterion of the IOTA lexicon and according to the risk of malignancy determined by the ADNEX model. The agreement between both methods for assigning the O-RADS group was estimated using weighted Kappa and the percentage of agreement. The sensitivity and specificity of both approaches were calculated. Results: 454 adnexal masses in 412 women were evaluated during the study period. There were 64 malignant masses. The agreement between the two approaches was moderate (Kappa: 0.47), and the percentage of agreement was 46%. Most disagreements occurred for the groups O-RADS 2 and 3 and for groups O-RADS 3 and 4. The sensitivity and specificity for O-RADS using the IOTA lexicon and O-RADS using the ADNEX model were 92.2% and 86.1%, and 85.9% and 87.4%, respectively. Conclusion: The diagnostic performance of O-RADS classification using the IOTA lexicon as opposed to the IOTA ADNEX model is similar. However, O-RADS group assignment varies significantly, depending on the use of the IOTA lexicon or the risk estimation using the ADNEX model. This fact might be clinically relevant and deserves further research.

## 1. Introduction

Adnexal masses are a common problem in gynecology. Fortunately, most of them are benign. However, ovarian cancer remains the most lethal gynecologic malignancy [1]. Accurate discrimination between benign and malignant adnexal masses is essential for adequate management. Adnexal lesions considered a low risk of malignancy could be managed expectantly or removed through minimally invasive techniques [2,3,4], whereas adnexal masses classified as a high risk of malignancy warrant further evaluation and eventually should be referred to a gynecologic oncology unit for adequate management [5,6].

Ultrasound is considered the first-line imaging technique for assessing adnexal masses, and no other imaging technique provides better diagnostic performance than TVS [7]. The subjective impression from an expert examiner using the so-called pattern recognition approach has been shown to be the best method for discriminating between benign and malignant adnexal masses [8]. However, this approach requires training, and it has been demonstrated that experience impacts diagnostic performance [9].

On the other hand, the proper description of ultrasound findings is essential in the process of communication between clinicians and sonographers. For this reason, in the year 2000, the International Ovarian Tumor Analysis group (IOTA) developed a consensus document about the lexicon to be used when describing adnexal masses as assessed by ultrasound that has become a standard worldwide [10]. Notwithstanding, most adnexal masses are usually assessed by non-expert examiners first. For this reason, IOTA has made significant efforts to develop several approaches, such as simple rules classification systems or logistic models, including the Assessment of Different NEoplasias in the Adnexa (ADNEX) model, in order for non-expert examiners to be able to achieve optimal performance discriminating benign from malignant adnexal masses [11,12,13]. 

However, ultrasound risk stratification remains a problem. In 2018, the American College of Radiology (ACR), jointly with IOTA, reported a white paper about the so-called Ovarian-Adnexal Report Data System (O-RADS) for classifying adnexal masses, aiming at providing a standardized lexicon that includes all pertinent descriptors and definitions of the characteristic sonographic appearance of normal ovaries and ovarian or other adnexal lesions [14]. 

In 2020, ACR reported a new consensus paper about the O-RADS classification developed, with the aim of guiding clinical management of adnexal masses based on the risk of malignancy stratification system [15]. The O-RADS system classifies adnexal masses into six categories for risk classification. These include: O-RADS 0, an incomplete evaluation; O-RADS 1, the physiologic category (normal premenopausal ovary); O-RADS 2, the almost certainly benign category (<1% risk of malignancy); O-RADS 3, lesions with a low risk of malignancy (1% to <10%); O-RADS 4, lesions with an intermediate risk of malignancy (10% to <50%); and O-RADS 5, lesions with a high risk of malignancy (≥50%). Risk group allocation can be done either by examiner interpretation of ultrasound findings using the IOTA lexicon or by estimating the risk using the ADNEX model. 

A recent meta-analysis has been reported showing that O-RADS classification offers a very good sensitivity, with moderate specificity [16]. However, a question to be answered is how much do the IOTA lexicon and ADNEX model agree on risk group allocation. Therefore, the aim of the present study is to assess the agreement of O-RADS classification when using the IOTA lexicon or the ADNEX model as the criteria for assigning the malignancy risk group and to compare the diagnostic performance of both approaches.

## 2. Materials and Methods

### 2.1. Study Design

This is a retrospective analysis of data prospectively collected, performed at two European centers between January 2019 and October 2021. Institutional Review Board approval was obtained prior to the start of the study.

### 2.2. Patients

All consecutive women diagnosed as having an adnexal mass and who underwent a transvaginal/transrectal ultrasound evaluation during the study period were eligible for the study. 

Inclusion criteria were as follows:Patients over 18 years of agePresented with at least one adnexal mass persistent for at least 3 monthsTransvaginal or transrectal ultrasound examination at one of the participating centersSurgery within 2 months after diagnosis, or at least 12 months of clinical and ultrasound follow-up if managed conservatively

Exclusion criteria were as follows: pregnancy at the time of the initial ultrasound evaluation or during the follow-up period; presence or history of any neoplastic disease; declining to undergo transvaginal or transrectal ultrasound; and less than 12 months of follow-up if expectant management was chosen.

### 2.3. Ultrasound Evaluation

As stated above, all patients ultimately included underwent transvaginal/transrectal ultrasound, according to a specific scanning protocol. This protocol was the same at both institutions. All examinations were performed by expert and non-expert examiners. The scanning protocol consisted of a detailed evaluation of the adnexal mass, according to IOTA recommendations [10]. The following ultrasound parameters were assessed: tumor size (three diameters in mm), type of lesion (unilocular, multilocular, unilocular-solid, multilocular-solid, or solid), external tumor contour, internal cyst wall (smooth or irregular), presence of papillary projections, number of papillary projections, size of papillary projections (maximum diameter), number of locules (in case of multilocular or multilocular-solid lesion), presence of solid component other than papillary projection, maximum size of solid component, color score (one to four), presence of acoustic shadows, and presence of ascites. In large adnexal masses, transabdominal ultrasound was also performed.

In case of the typical appearance of ovarian endometrioma, dermoid cyst, simple cyst, hemorrhagic cyst, or any other unilocular cyst, the examiner had to record this finding. All examinations were recorded in video clips and hardcopy images, and a written report elaborated. 

### 2.4. Clinical Management

Clinical management was decided based on the examiner’s subjective impression using so-called pattern recognition, taking also into account the patient’s clinical features and symptoms. Note that management was not based on O-RADS classification.

Asymptomatic benign-appearing masses with less than 10 cm maximum size were submitted to serial follow-up (every six months), unless the patient asked for surgery. Asymptomatic benign-appearing masses with more than 10 cm maximum size and symptomatic benign-appearing masses were submitted to surgery by the general gynecologist.

A definitive histological diagnosis was obtained in the case of surgical removal. As stated above, only women with at least 12 months of follow-up in the case of conservative management were included in the study. Masses with a suspicious appearance were referred to the Gynecologic Oncology Unit for management. 

### 2.5. O-RADS Classification Assessment

Retrospectively, one single examiner in each center reviewed all written reports, videos, and hardcopies of each adnexal mass for classifying the adnexal mass using O-RADS classification according to the IOTA lexicon (Table 1). 

The ADNEX model was also used to estimate the risk of malignancy, and the O-RADS group was assigned according to the result of the ADNEX model (Table 2).

### 2.6. Statistical Analysis

Continuous variables were expressed as a mean with standard distribution (SD) and range or a median with interquartile range (IQR) and range, depending on the data distribution. Categorical variables were expressed as a number with percentage. Categorical data were compared using the chi-square test. Continuous data were compared using one-way ANOVA or the U Mann–Whitney test, depending on the data distribution. The Kolmogorov–Smirnov test was used to test the data distribution.

To estimate the diagnostic performance of O-RADS classification using the IOTA lexicon and using the IOTA ADENX model, we calculated the sensitivity, specificity, positive likelihood ratio (+LR), and negative likelihood ratio (-LR), for each approach. Sensitivity and specificity were compared using the McNemar test. For calculations, O-RADS 2 and 3 lesions were considered benign, and O-RADS 4 and 5 lesions were considered malignant.

Agreement for O-RADS risk group assignment between both approaches was estimated by calculating the weighted kappa index and the percentage of agreement in classifying the mass as benign, inconclusive, or malignant. A kappa value of <0.20 indicates poor agreement, 0.21–0.40 indicates fair agreement, 0.41–0.60 indicates moderate agreement, 0.61–0.80 indicates good agreement, and 0.81–1.00 indicates very good agreement. GraphPad QuickCalcs software was used to calculate the kappa and weighted indices (GraphPad Software Inc., La Jolla, CA, USA). Power and sample size estimations were not performed. 

We used the Standards for Reporting Diagnostic Accuracy guidelines (STARD) [17] and the Guidelines for Reporting Reliability and Agreement Studies (GRRAS) [18].

## 3. Results

In total, 454 adnexal masses in 412 women (mean age: 48.3 years, SD: 14.4, range: 18 to 87 years old) were evaluated during the study period. A total of 231 women (56.1%) were premenopausal, and 181 women (43.9%) were postmenopausal.

Moreover, 240 masses (52.9%) were evaluated at center A, and 214 masses (47.1%) were evaluated at center B. In addition, 230 masses (50.7%) were managed conservatively, and 224 (49.3%) were removed surgically. There were 64 (14.1%) malignant masses. None of the patients with an adnexal mass managed conservatively has developed ovarian cancer. Therefore, the total number of benign masses was 390 (85.9%).

The median tumor maximum size was 43.0 mm (IQR: 39), ranging from 5.0 mm to 200.0 mm. The examiner’s subjective impression diagnosis was benign in 369 cases (81.3%) and malignant in 85 cases (18.7%). Patients’ characteristics were similar in both centers. However, malignant cases were more frequent in one center than the other (Table 3). 

The O-RADS classification of the cases using the IOTA lexicon was as follows: O-RADS 0: 0 cases, O-RADS 1: 0 cases, O-RADS 2: 250 cases (55.1%), O-RADS 3: 91 cases (20.0%), O-RADS 4: 70 cases (15.4%), and O-RADS 5: 43 cases (9.5%). The median ADNEX model risk result in the whole series was 3.1% (IQR: 5.4), ranging from 0.1% to 99.3%. The O-RADS classification of the cases using the ADNEX model risk estimation was as follows: O-RADS 0: 0 cases, O-RADS 1: 0 cases, O-RADS 2: 74 cases (16.3%), O-RADS 3: 281 cases (61.9%), O-RADS 4: 61 cases (13.4%), and O-RADS 5: 38 cases (8.4%). Table 4 shows the O-RADS distribution in both centers, according to the lexicon and ADNEX. O-RADS 4 cases using both approaches were more frequent in center B.

Table 5 shows the distribution of O-RADS groups, according to classification using the IOTA lexicon and the ADNEX model in the whole series. 

The agreement between the two approaches for assigning a specific risk group was moderate (Weighted Kappa: 0.47, 95%CI: 0.42–0.53), with a percentage of agreement of 46% (95%CI: 42–51%). Most disagreements occurred for the groups O-RADS 2 and 3 (73.2% of the cases classified as O-RADS 2 by the IOTA lexicon are classified as ORADS-3 by ADNEX) and for the groups O-RADS 3 and 4 (30.0% of the cases classified as O-RADS 4 by the IOTA lexicon are classified as O-RADS 3 by ADNEX).

However, grouping O-RADS 2 and O-RADS 3 as very low- or low-risk tumors and O-RADS 4 and O-RADS 5 as intermediate- or high-risk tumors, the agreement between both approaches (the IOTA lexicon and the ADNEX model) was good (Kappa index: 0.75, 95%CI: 0.67–0.82), with a percentage of agreement of 91% (95%CI: 87–93%). We did not find statistically significant differences in the diagnostic performance of either approach (McNemar test. *p* = 0.221). The sensitivity, specificity, +LR, and −LR for O-RADS classification using the IOTA lexicon were 92.2% (95%CI: 83.0–96.6%), 86.1% (95%CI: 82.3–89.2%), 6.6 (95%CI: 5.1–8.6), and 0.09 (95%CI: 0.04–0.21), respectively. The sensitivity, specificity, +LR, and −LR for O-RADS classification using the ADNEX model were 85.9% (95%CI: 75.4–92.4%), 87.4% (95%CI: 83.8–90.4%), 6.8 (95%CI: 5.2–9.0), and 0.16 (95%CI: 0.09–0.29), respectively (Table 6 and Table 7).

## 4. Discussion

When using the O-RADS ultrasound-based risk estimation for adnexal masses proposed by the American College of Radiology, the risk group categorization for a specific lesion may be done by using two strategies, namely, the IOTA lexicon or the IOTA ADNEX model [15]. In this seminal paper, the ACR did not state which approach should be used or whether one approach could be preferable to the other. 

The IOTA lexicon approach is based on the application of IOTA ultrasound descriptors, according to IOTA terms and definitions, considering mainly the type of lesion (unilocular, multilocular, unilocular-solid, multilocular-solid, or solid), internal contour of the cyst’s wall, tumor size, tumor external contour, and vascular color score [10]. No other features, such as clinical data (for example, patient’s age) or biochemical parameters (for example, CA-125 serum levels), are considered. The risk estimation for each O-RADS category was derived from the prevalence of malignancies observed in large IOTA phase 1–3 studies (5905 women) [11,19,20,21]. The prevalence of malignancy in that database for tumors classified as O-RADS 2 (N = 1425 patients), O-RADS 3 (N = 945 patients), O-RADS 4 (N = 1734 patients), and O-RADS 5 (N = 1774 patients) was 0.5%, 3.6%, 29.8%, and 77.5%, respectively. 

The IOTA ADNEX model is a logistic model developed from data derived from IOTA phase 1, 2, and 2b studies (3506 women) (13). This model includes six ultrasound variables (tumor maximum size, maximum size of the solid component, number of locules, number of papillary projections, ascites, and acoustic shadows), two clinical variables (patient’s age and type of center—oncologic or not oncologic), and one biochemical variable (absolute value of the CA-125 serum tumor marker). The risk estimation for each O-RADS category is derived from the result of applying a mathematical formula. This model provides not only a dichotomic estimation (risk of the lesion being benign or malignant), but also the risk of being a borderline tumor, early-stage invasive ovarian cancer, advanced-stage ovarian cancer, or metastatic tumor to the ovary. It is important to note that the CA-125 value can be used, or not, for calculating the risk of malignancy. However, the CA-125 value is essential for calculating the risk of the malignant tumor type (borderline tumor, early stage invasive ovarian cancer, advanced stage ovarian cancer, or metastatic tumor to the ovary).

It is clear that both strategies differ on how they estimate the risk of malignancy for a given adnexal mass in order to assign the risk group. For this reason, we wondered how much they agree. We have observed that the agreement is moderate for assigning the specific risk group (O-RADS 2, 3, 4, or 5), with more than 50% of cases allocated in a different risk group, depending on the strategy used. 

In terms of diagnostic performance, we have observed that both approaches offer good sensitivity and specificity. These results are in agreement with a recent meta-analysis regarding O-RADS classification performed by our group [16]. We found 11 studies reported from 2021 to 2022 assessing the role of O-RADS classification for discriminating between benign and malignant adnexal masses. The pooled sensitivity and specificity were 97% and 77%, respectively. All the studies but one were retrospective external validations. Furthermore, all of them used the IOTA lexicon to assign the risk group; not one used the ADNEX model to assign the risk group. By the study’s design, no study compared the group assignment using the IOTA lexicon or the ADNEX model. The study from Hiett and colleagues compared O-RADS classification using the IOTA lexicon with the ADNEX model (not O-RADS risk stratification using the ADNEX model) [22]. They found that the sensitivity and specificity for O-RADS using the IOTA lexicon was 100% and 46.4%, respectively. However, these figures for the ADNEX model using a 10% cut-off were 97.5% and 63.6%, respectively. Somehow, these authors made a comparison similar to ours.

Nonetheless, we did not find differences regarding the diagnostic performance for both approaches; yet, we do think that our results can be relevant from the clinical point of view when considering risk group assignment since the ACR O-RADS classification system also proposed an algorithm for patient management according to risk estimation [15]. Patients with an O-RADS 2 (almost certainly benign) lesion could be managed expectantly, or they could be referred to an expert examiner evaluation or magnetic resonance assessment if concerns exist. O-RADS 3 (low risk) masses could be managed by a general gynecologist, or they could be referred to an expert examiner evaluation or magnetic resonance assessment if concerns exist. O-RADS 4 (intermediate risk) lesions could be managed by a general gynecologist with a gyne-oncologist consultation, or they could be referred to a Gynecologic-Oncology Unit. Like with O-RADS 2 and 3, in the case of concern, the patient could be assessed by an expert sonographer or undergo a magnetic resonance imaging evaluation. Finally, O-RADS 5 (high risk) masses should be referred to a Gynecologic-Oncology Unit.

In fact, a recent consensus on the pre-operative assessment of adnexal masses released jointly by the European Society of Gynecologic Oncology (ESGO), the International Society of Ultrasound in Obstetrics and Gynecology (ISUOG), the European Society of Gynecologic Endoscopy (ESGE), and IOTA states that, in spite of the O-RADS classification system not having been validated yet, it should be used determine patient management [23]. According to this consensus, masses classified as O-RADS 2 should be managed with follow-up, and masses classified as O-RADS 3 should be submitted to surgery in general gynecology units. However, masses classified as O-RADS 4 or 5 should be referred to specialized gynecologic oncology centers. Our data show that 73% of the cases classified as O-RADS 2 by the IOTA lexicon would be classified as O-RADS 3 by ADNEX. Therefore, a clear disagreement on case management would occur (most women submitted to follow-up using the IOTA lexicon strategy would be submitted to surgery if the ADNEX model is used). Similarly, 30% of the O-RADS 4 cases, as assigned by the IOTA lexicon strategy, which should be referred to specialized oncologic centers, would be classified as O-RADS 3 if the ADNEX model is used, and they would be submitted to surgery in general gynecologic units. 

Our findings are even more relevant when reproducibility is considered. Some recent studies have shown a significant inter-reader variability among observers when the IOTA lexicon is used [24], especially in the case of less experienced examiners [25]. Furthermore, it seems that training is needed to achieve good diagnostic performance using this classification system [26]. This could imply that using the ADNEX model could be preferable.

The main strength of our study is that, to the best of our knowledge, this is the first reported study addressing this issue. However, our study has limitations. We retrospectively stratified adnexal masses, both for the IOTA lexicon and the ADNEX model strategy. Therefore, in spite of only including patients with all data available, a risk of bias in patient selection could exist. Additionally, the same examiner in each center reviewed all the images and stratified the lesions according to the IOTA lexicon and the ADNEX model strategies. This fact also poses a risk of bias, and no inter-observer reproducibility assessment could be performed. Finally, we did not estimate sample size nor calculate statistical power.

Interestingly, a similar classification has been developed for magnetic resonance imaging, which was initially called ADNEX-MRI [27]. After proper prospective evaluation [28], the American College of Radiologists adopted this classification, the so-called O-RADS MRI [29]. This classification includes five risk categories and is based on the evaluation of the T1- and T2-weighted signal intensity, diffusion-weighted imaging properties, and lesion contrast agent uptake on dynamic contrast-enhanced (DCE) images. A recent meta-analysis comprising 12 studies and 4520 adnexal masses has shown that this classification has a high diagnostic performance in this clinical scenario, with a pooled sensitivity and specificity of 92% and 91%, respectively [30]. Albeit this MRI-based classification was developed for discriminating benign form malignant adnexal masses classified as indeterminate by ultrasound, it can be used for any adnexal mass. 

However, there are no studies assessing the O-RADS MRI system on non-indeterminate adnexal masses. Notwithstanding, another recent meta-analysis comparing O-RADS ultrasound (15 studies) and O-RADS MRI (12 studies) found that both systems had similar sensitivity (95% for both), but O-RADS MRI had better specificity (90%) as compared to O-RADS ultrasound (82%) [31]. However, it should be noted that most O-RADS MRI studies focused on indeterminate adnexal masses as assessed by ultrasound, whereas all O-RADS ultrasound studies focused on any type of adnexal mass. Therefore, a risk of selection bias exists in this meta-analysis. 

## 5. Conclusions

In conclusion, we have observed that agreement is moderate for assigning specific risk groups and variates according to the strategy used. In spite of the limitations mentioned above, we do think that our results deserve attention. The American College of Radiology should provide some guidance regarding which strategy (IOTA lexicon or ADNEX model) should be preferable to use for assigning O-RADS risk groups. Using one or the other might influence the clinical management of patients diagnosed as having an adnexal mass. Notwithstanding, we do think that prospective studies are needed to confirm our results.

## Figures and Tables

**Table 1 diagnostics-13-00673-t001:** O-RADS classification according to IOTA lexicon.

O-RADS Group	Ultrasound Descriptors	Risk of Malignancy
O-RADS 0	Incomplete evaluation	Not stated
O-RADS 1	Normal premenopausal ovary	0%
O-RADS 2	Classic hemorrhagic cyst ≥ 5 cm to <10 cm Classic dermoid cyst < 10 cm Classic endometrioma < 10 cm Unilocular smooth cyst ≤ 3 cm Other unilocular smooth cyst ≥ 3 cm to <10 cm	<1%
O-RADS 3	Unilocular smooth ≥ 10 cm Unilocular irregular wall Multilocular smooth CS 1–3 < 10 cm Solid smooth CS 1	1% to <10%
O-RADS 4	Multilocular smooth ≥ 10 cm CS 1–3 Multilocular smooth CS 4 Multilocular irregular Unilocular-solid no papillary projection Unilocular-solid 1–3 papillary projections Multilocular-solid CS 1–2 Solid smooth CS 2–3	10% to <50%
O-RADS 5	Unilocular-solid with ≥ 4 papillary projections Multilocular-solid CS 3–4 Solid smooth CS 4 Solid irregular Ascites or metastases	50% to 100%

CS: Color score.

**Table 2 diagnostics-13-00673-t002:** Parameters used in the ADNEX model.

ADNEX Parameter	Value
Patient’s age	Numerical value
Oncology center?	No/Yes
Maximal diameter of the lesion	Numerical value
Maximal diameter of the largest solid part	Numerical value
More than 10 locules?	No/Yes
Number of papillary projections	None/One/Two/Three/More than three
Acoustic shadows present?	No/Yes
Ascites (fluid outside pelvis present)?	No/Yes
CA-125 (U/mL)	Numerical value

CA-125 serum levels were assessed only in patients with suspicious masses according to O-RADS classification using the IOTA lexicon (O-RADS 4 and 5). Thus, CA-125 was not used for ADNEX model calculation.

**Table 3 diagnostics-13-00673-t003:** Basic patients and tumor characteristics in the two participating centers.

	Center A	Center B	*p*-Value
Patient’s age (years) *	47.6 (13.8)	49.0 (14.9)	0.329
Postmenopausal women **	81 (40.8)	100 (46.7)	0.274
Maximum tumor size (mm) †	36.0 (29.0)	50.0 (45.0)	<0.001
Malignant cases **	23 (9.3)	41 (19.2)	0.003

* Expressed as mean with standard deviation in parentheses. ** Expressed as number with percentage in parentheses. † Expressed as median with interquartile range in parentheses.

**Table 4 diagnostics-13-00673-t004:** Distribution of cases according to O-RADS strategy used in each participating center.

	Center A	Center B
IOTA Lexicon		
O-RADS 2	154 (64.2%)	96 (44.9%)
O-RADS 3	46 (19.2%)	45 (21.0%)
O-RADS 4	16 (6.7%)	54 (25.2%)
O-RADS 5	24 (10.0%)	19 (8.9%)
ADNEX model		
O-RADS 2	28 (11.7%)	46 (21.5%)
O-RADS 3	173 (72.1%)	108 (50.5%)
O-RADS 4	25 (10.4%)	36 (16.8%)
O-RADS 5	14 (5.8%)	24 (11.2%)

**Table 5 diagnostics-13-00673-t005:** Distribution of cases of O-RADS risk group according to the strategy used.

		O-RADS According to IOTA Lexicon			
		O-RADS 2	O-RADS 3	O-RADS 4	O-RADS 5
O-RADS according to ADNEX model	O-RADS 2	64	183	3	0
	O-RADS 3	7	76	8	0
	O-RADS 4	3	21	39	7
	O-RADS 5	0	1	11	31

**Table 6 diagnostics-13-00673-t006:** Distribution of cases according to O-RADS classification using IOTA lexicon.

	Reference Standard		
	Benign	Malignant	Total
O-RADS 2/3	336	5	341
O-RADS 4/5	54	59	113
Total	390	64	454

**Table 7 diagnostics-13-00673-t007:** Distribution of cases according to O-RADS classification using ADNEX model.

	Reference Standard		
	Benign	Malignant	Total
O-RADS 2/3	341	9	350
O-RADS 4/5	49	55	104
Total	390	64	454

## Data Availability

Data are available upon reasonable request.

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
