# Peer review of "O-RADS Classification for Ultrasound Assessment of Adnexal Masses: Agreement between IOTA Lexicon and ADNEX Model for Assigning Risk Group"

_diagnostics, 2023, doi:10.3390/diagnostics13040673_

Round 1
Reviewer 1 Report
The authors conducted the proper collection of data
Good English level. Recommend publication.
Author Response
Question: The authors conducted the proper collection of data. Good English level. Recommend publication.
Answer: Thanks for this comment. We appreciate
Reviewer 2 Report
Dear Author,
It’s an interesting article.
Adrenal masses remain a challenge in terms of diagnosis and treatment for both gynecologist and, endocrinologist. I consider that the article meets the criteria for publication.
Kind regards,
Author Response
Question: Dear Author, It’s an interesting article. Adnexal masses remain a challenge in terms of diagnosis and treatment for both gynecologist and, endocrinologist. I consider that the article meets the criteria for publication. Kind regards.
Answer: Thanks for this comment. We appreciate
Reviewer 3 Report
N/A
Author Response
No comment
Thanks for reviewing
Reviewer 4 Report
MRI is currently use in daily practice explore adnexal masses.
MRI ORADS score is highly effective.
Not any words is discussed about it in your manuscript
Author Response
Question: MRI is currently use in daily practice explore adnexal masses. MRI ORADS score is highly effective. Not any words is discussed about it in your manuscript
Answer: We add some sentences in Discussion
Reviewer 5 Report
I read with great interest the Manuscript titled "O-RADS classification for ultrasound assessment of adnexal masses: agreement between IOTA lexicon and ADNEX model for assigning risk group" which falls within the aim of the Journal.
In my honest opinion, the topic is interesting enough to attract the readers’ attention. Methodology is accurate and conclusions are supported by the data analysis. Nevertheless, authors should clarify some point and improve the discussion citing relevant and novel key articles about the topic.
- I suggest another round of language revision, in order to correct few typos and improve readability.
- Discussions can be expanded and improved by citing relevant articles (I suggest authors to read and insert in references the following article PMID: 35343352).
For these reasons, I recommend the publication of the article, pending few minor revisions.
Author Response
Question: I read with great interest the Manuscript titled "O-RADS classification for ultrasound assessment of adnexal masses: agreement between IOTA lexicon and ADNEX model for assigning risk group" which falls within the aim of the Journal. In my honest opinion, the topic is interesting enough to attract the readers’ attention. Methodology is accurate and conclusions are supported by the data analysis. Nevertheless, authors should clarify some point and improve the discussion citing relevant and novel key articles about the topic.
Answer: thanks for this comment
Question: I suggest another round of language revision, in order to correct few typos and improve readability.
Answer: English language revised
Question: Discussions can be expanded and improved by citing relevant articles (I suggest authors to read and insert in references the following article PMID: 35343352).
Answer: Dear Reviewer. We found that PMID 35343352 refers to this article ((2022) Predictors of Pain Development after Laparoscopic Adnexectomy: A Still Open Challenge, Journal of Investigative Surgery, 35:6, 1392-1393, DOI: 10.1080/08941939.2022.2056274) , which is a comment on other article ( (2022) Surgical Determinants of Post Operative Pain in Patients Undergoing Laparoscopic Adnexectomy, Journal of Investigative Surgery, 35:6, 1386-1391, DOI: 10.1080/08941939.2022.2045395) . We are not sure whether you actually refer to it since these papers refer to pain after surgery and not to differential diagnosis of adnexal masses. It seems they are not closely related to the topic of our research. Please, could you confirm you refer to these manuscripts?
We have made no change in the manuscript.
Question: For these reasons, I recommend the publication of the article, pending few minor revisions.
Answer: Thanks for this comment